# Current Evidence in the Systemic Treatment of Brain Metastases from Breast Cancer and Future Perspectives on New Drugs, Combinations and Administration Routes: A Narrative Review

**DOI:** 10.3390/cancers16244164

**Published:** 2024-12-13

**Authors:** Ornella Garrone, Fiorella Ruatta, Carmen Giusy Rea, Nerina Denaro, Michele Ghidini, Carolina Cauchi, Claudia Bareggi, Barbara Galassi, Marco C. Merlano, Roberto Rosenfeld

**Affiliations:** 1Department of Medical Oncology, Fondazione IRCCS Ca’ Granda Ospedale Maggiore Policlinico, 20122 Milan, Italy; fiorella.ruatta@policlinico.mi.it (F.R.); carmen.rea@policlinico.mi.it (C.G.R.); nerina.denaro@policlinico.mi.it (N.D.); michele.ghidini@policlinico.mi.it (M.G.); carolina.cauchi@policlinico.mi.it (C.C.); claudia.bareggi@policlinico.mi.it (C.B.); barbara.galassi@policlinico.mi.it (B.G.); roberto.rosenfeld88@gmail.com (R.R.); 2Scientific Direction, Candiolo Cancer Institute, FPO-IRCCS Candiolo, 10060 Torino, Italy; mcmerlano@gmail.com

**Keywords:** breast cancer, brain metastases, systemic treatment

## Abstract

Brain metastases represent an unmet medical need in the treatment of breast cancer. Notwithstanding significant achievements in the management of both early and advanced settings, the spread of the disease to the brain is challenging. A multidisciplinary approach allows selection of the most appropriate trajectory. Patients with HER2-positive and triple-negative breast disease are at major risk to develop brain deposits. Local treatment, surgery and radiotherapy are useful in the treatment of brain metastases; however, systemic therapy, small molecules and antibody drug conjugates can be successfully used. In this review, we summarize the systemic treatment landscape of brain metastases from breast cancer to shed light on the most favorable strategies.

## 1. Introduction

Breast cancer (BC) is the most frequently diagnosed neoplasm all over the world and the second leading cause of cancer death in women [1]. Early detection of the disease with screening and the widespread use of adjuvant therapy have significantly improved the prognosis [2]. However, metastatic disease remains incurable. Notwithstanding, brain metastases (BMs) have been considered a rare and late event for decades; as women with stage IV disease live longer, they are at a higher risk for developing central nervous system deposits over time [3,4,5,6,7,8,9,10,11,12]. Metastatic spread to the brain is associated with a poor prognosis and affects quality of life. Therefore, it represents a major unmet need in patients with BC.

Patients with human epidermal growth factor receptor 2 (HER2)-positive or triple-negative breast cancer (TNBC) are at a higher risk of developing BM than patients with luminal disease (reviewed in [13]). Data from retrospective analyses showed that roughly half of patients with the most aggressive subtypes experience progression to the brain and eventually die [3,4,5,6,7,10,11,12].

The development of new therapeutic agents such as small molecules and antibody–drug conjugates (ADCs) have contributed to the improvement of patients’ prognosis with BMs from BC [14,15,16,17,18,19,20]. Cyclin-dependent kinase 4/6 inhibitors (CDK4/6i), such as abemaciclib, have demonstrated activity for the treatment of BMs in luminal-B BC patients [21].

However, patients with known brain involvement are generally excluded from randomized trials, making difficult to have sufficient evidence regarding the activity of systemic therapy. In addition, subtype changes have been described in up to 30% of BC patients developing brain deposits [22,23,24,25,26], and systemic therapy is responsible for clonal selection of the neoplastic cells contributing to the heterogeneity of metastatic spread [27].

The aim of this narrative review is to shed light on the prognostic and molecular factors of brain metastases from breast cancer along with the breakthroughs in systemic therapy, as well as future perspectives.

## 2. Materials and Methods

### 2.1. Aim and Categories

This narrative review aims to gather new and old data in order to clarify the complex mechanisms of BM from BC with subdivision of the data into three main categories (Figure 1): (i) molecular mechanisms identified from preclinical in vitro and in vivo models, (ii) positive clinical trials investigating the efficacy of available drugs on BMs and (iii) future perspectives, focusing on the most relevant past failed clinical trials and the promising ongoing clinical trials on oncological drugs directed against BMs. For this purpose, we conducted an extensive literature search, querying PUBMED, the most used database worldwide, from 2000 to August 2024, gathering the large part of indexed journals. All the queries and the research items were defined and developed by four experienced authors, dividing the tasks according to the authors’ expertise.

### 2.2. Research Strings and Inclusion/Exclusion Criteria

A research question was formulated in order to define the population, criteria, and three main categories for querying the databases. For this purpose, MeSH terms were used, as well as keywords conjugated by Boolean connectors (AND, OR, NOT) using strings of keywords such as “Breast [title]”, “cancer OR tumor OR neoplasia OR tumour OR neoplasm”, “metastases[All Fields]”, “brain[title] OR encephalic[title] OR head[title] OR leptomeningeal[title]” in order to define the population and the matter of the research. Moreover, aiming to retrieve articles for the main categories, we used additional MeSH terms in combined keywords such as “randomized”, “clinical” and “trial” OR “observational” and “study” OR “translational” and “study” OR “mice” OR “mouse” OR “animal”. Eventually, we also adopted negative Boolean connectors (NOT) aiming to significantly reduce inappropriate articles focusing on primary brain tumors (“NOT glioblastoma NOT glioma”).

All the authors were involved in the choice of the keywords for research query construction. From results, we included full-length, English-written, original articles focused on BMs from BC and within the categories previously mentioned. Furthermore, the following articles were excluded: (i) papers written in non-English languages, (ii) abstract-only articles, (iii) not fully published studies, (iv) case reports/series and (v) reviews or letters/commentaries (Figure 1-S2). An exception was made for congress abstracts not yet followed by full-length paper publication for new studies that deserved to be discussed. In addition, studies focusing on non-pertinent arguments were excluded, such as (a) pharmacokinetics and physiology or pathophysiology studies on blood–brain barrier (BBB) permeability, (b) diagnostics studies, (c) innovative animal or xenograft models, (d) non-pharmaceutical interventions and (e) radiotherapy interventions without a drug in combination (Figure 1-S3).

### 2.3. Selection Procedures of the Sources of Evidence

We started querying the database of PubMed from 2000 until 1 August 2024. The first step focused on gathering titles with the specific inclusion and exclusion criteria mentioned above (Figure 1). The collected articles were gathered in reference management software for the further evaluations, and four authors were independently involved in the process of selection of the articles. The procedure was pursued in a first round evaluating titles (Figure 1), applying inclusion/exclusion criteria aiming to drop duplicates, reviews and old articles. At this point, a second round was started, focusing on screening the abstracts for congruency with the inclusion criteria and verifying the consistency of arguments; otherwise, if the inclusion criteria were deemed to be marginally met, the study was discarded. Once the process was terminated, a cross-over evaluation of the gathered articles was made by the four authors, who checked for the correctness of the included studies. In cases of disagreements, a third author revised the cases, settling the final decision. Eventually, the full texts of remaining selected articles were obtained to confirm eligibility and the presence of relevant data for the review.

## 3. Incidence and Risk Factors

The risk of developing BMs in metastatic breast cancer (mBC) patients is roughly 30–50% [28]. In a French retrospective analysis of 16703 mBC patients, the incidence of BM, after a median follow-up of 43 months, was as high as 25% [29].

The most common factors associated with an increased likelihood of developing BMs include disease stage, younger age at diagnosis, shorter disease-free interval, the presence of extracerebral disease (lymphatic, pulmonary and hepatic), the number of metastatic sites, a high histological grade, BRCA1 phenotype and molecular subtype. [6,30,31,32,33,34,35]. In particular, triple-negative and HER2-positive disease are the molecular subtypes most likely associated with brain metastases. Indeed, BMs from TNBC are an early event and frequently associated with extracerebral disease, leading to short survival (median OS 6 months) reviewed in [36].

## 4. Biomolecular Features

Effective treatment of BMs basically requires three main factors: the availability of active treatments for the primary tumor, comprehension of the tumor molecular mechanisms underlying the formation and survival of BMs and knowledge about the unique microenvironment of the BMs. Among the tumors likely to develop BMs [37], breast cancer has recently benefitted from highly effective drugs, such as immunotherapy, tucatinib and trastuzumab deruxtecan, thus improving its outcome.

However, most breast cancer patients with BMs still experience poor survival [38], suggesting that the available effective drugs targeting the primary tumor are not sufficient to control BMs. Interestingly, data from The Cancer Genome Atlas (TCGA) reported in a large study considering 33 different tumor types indicate that metastases and the tumor of origin share the same mutational landscape [39].

Other data derived from TCGA show that different cancer types share some gene expression patterns associated with metastasis, although each type of cancer has a different metastasis signature. The gene expression patterns shared across different tumor types included genes related to the stress response, oxidation–reduction process, protein ubiquitination, DNA repair and receptor activity [40]. This finding suggests that knowledge of the molecular mechanisms leading to the survival and growth of cancer cells within the brain may help identify actionable targets for new therapies broadly directed against BMs or focusing on single-cancer type-specific BMs.

The last aspect that needed to be explored was the interplay between the brain microenvironment and cancer cells, that is, the BM microenvironment.

Microglia, bone marrow-derived macrophages and astrocytes cooperate efficiently to defend brain tissue. It is hypothesized that 99% of cancer cells penetrating the brain do not survive [41]. However, once a cancer cell survives, it becomes able to modulate the metastatic niche to allow the growth of cancer cells and their survival. Under the effect of cancer cells, microglia reduce the expression of iNOS and tumor necrosis factor (TNF)-α and suppress cytotoxic activity [42]. Breast cancer cells (BCCs) can also manipulate microglia through the expression of neuronal proteins such as neurotrophin (NT)-3, a modulator of microglial activation, on the cell surface. NT-3 promotes the development of clinically detectable lesions in the brain and contributes to the switch from the mesenchymal to the epithelial phenotype, allowing the survival of cancer cells in the brain. Additionally, in a mouse model, NT-3 favored the proliferation of BCCs in the brain [43]. Immunomodulation by NT-3 prevents microglial activation [44]. It was observed that low levels of microglial activation favor metastatic growth, whereas high levels of activation produce antitumor effects.

Moreover, in vitro studies highlighted that microglia may also promote brain invasion. Gene expression analyses of microglia (co-cultured with cancer cells) identified the TLR, WNT/β-catenin, CXCL-12/CXCR4, PI3K and CCL2/CCR2 pathways as the most affected. These molecular signals are K-factors driving microglia–cancer cell cross-talk, and their inhibition may have a positive impact on the control of brain metastatization from BC. For instance, microglia-induced protumoral activity was completely abrogated by the WNT antagonist DKK-2 [45]; TLR-4 agonists were able to restore the anticancer activity of microglia [46], and treatment with a pan-PI3K Class I inhibitor, buparlisib (BKM120), reduced metastasis-promoting activity [47].

Similar to microglia, astrocytes are a double-edged sword. On one hand, they cooperate with endothelial cells to counteract cancer cells infiltration. On the other hand, astrocytes induce BCCs to produce high amounts of IL-1β and TNF-α, which activate NOTCH signaling, promoting the stemness of cancer stem cells and ultimately tumor growth [48]. This point will be further explored later.

In turn, IL-1β and TNF-α promote the production of TGF-β by astrocytes [49]. Furthermore, astrocytes produce extracellular matrix-degrading factors that favor invasion and migration of BCCs within the brain [50].

Recently, it was observed that TNBC metastatic cells expressed N-methyl-D-aspartate (NMDA) receptors on their surfaces, although the tumoral cells and the niche did not secrete sufficient glutamate for this pathway’s stimulation. However, BM cells seem to interact with neurons and synapses rather than simply disrupting them. Indeed, it was observed that malignant cells can replace the position of astrocytes and mimic their synaptic processes in a spatial development analogous to a tripartite synapse with a neuron–neuron–tumor contact. In this setting, Zheng et al. proved that an increase in glutamate production from neurons led to stimulation of NMDA receptors on BM cells, enhancing tumoral growth, invasion and colonization [51].

Growing interest focuses on the role of pericytes in supporting cancer metastasis in the brain. Pericytes produce insulin-like growth factor 2 (IGF2), which supports the proliferation of mammary carcinoma but not other cancers [52]. Indeed, the inhibition of IGF2 signaling reduced the size of BMs in mice inoculated with TNBC cells. These observations support the hypothesis that brain pericytes play a significant pro-metastatic role, at least in BC [52].

Among the three main BC subtypes, TNBC is the one most associated with BMs. This subtype, more than the others, is able to disrupt the blood–brain barrier (BBB), facilitating the diffusion of cancer cells to the brain [53,54] via upregulation of surface catepsin-S expression, which induces lysis of tight junctions of the BBB [55]. This characteristic contributes to explaining the high frequency and rapid onset of BMs in TNBC patients [36]. However, other aspects may also explain the aggressivity of TNBC BMs. Among them, it has been shown that activated astrocytes (see above) stimulate stem cells through the activation of NOTCH signaling. Accordingly, TNBC harbors the highest rate of tumoral stem cells among all the BC subtypes, and more generally, it shows molecular and transcriptional activity characteristics typical of cancer stem cells.

In addition, Dionísio et al., in a translational study, observed not only a significant increase in stem cell frequency with tropism to the brain but also a clear difference in these cells’ phenotype, resulting in higher cohesion and an epithelial-like phenotype [56].

Therefore, astrocytes may support the highest aggressivity of TNBC within brain tissue by activating NOTCH [48], and NOTCH might be a good target for specific treatment of TNBC BMs.

All the above data highlight the importance of deepening the knowledge of the molecular features and the interaction that occurs in the microenvironment of BC-associated BMs, as they could allow the identification of targets for future treatments.

## 5. Systemic Treatment

### 5.1. HER2-Positive Breast Cancer

Nearly 50% of patients with advanced HER2-positive  BC will eventually develop BMs, even in cases of absent or stable extracranial disease [13,57]. Systemic treatment of BMs is based on several anti-HER2 drugs with distinct mechanisms of action, such as monoclonal antibodies (trastuzumab and pertuzumab), tyrosine kinase inhibitors (TKIs; lapatinib, neratinib and tucatinib) and antibody-drug conjugates (trastuzumab emtansine and trastuzumab deruxtecan).

The combination of trastuzumab and pertuzumab with a taxane represents the mainstay of first-line treatment in HER2-positive mBC.

Pertuzumab acts synergically with trastuzumab by binding with the HER2 extracellular domain. However, trastuzumab alone cannot pass the BBB [12].

The CLEOPATRA trial showed meaningful prolongation of both progression-free survival (PFS) and overall survival (OS) in HER2-positive mBC patients treated with the combination of dual blockade and docetaxel compared to placebo, trastuzumab and docetaxel [58]. Notably, although the trial did not include patients with BMs, there is evidence that dual blockade delays the onset of brain disease by 3 months (from 12 to 15 months) [59].

In a real-world registry for BMs, dual blockade with trastuzumab plus pertuzumab yielded longer OS (nearly 44 months) than other HER2-targeted therapies, including trastuzumab alone, trastuzumab plus lapatinib and lapatinib alone [60].

Considering TKIs, they are commonly used in later lines of treatment, but interestingly, they are able to efficiently cross the BBB. Indeed, the first TKI approved in this setting was lapatinib, a reversible inhibitor of both epidermal growth factor receptor (EGFR) and HER2 [15].

In the LANDSCAPE trial, patients with previously untreated BMs who were not suitable for surgical resection received the combination of lapatinib plus capecitabine. The combination resulted in a central nervous system (CNS) objective volumetric response in nearly 66% of the population and a median time to CNS progression of 5.5 months [16]. Diarrhea, hand–foot syndrome and fatigue were frequently associated with this treatment.

However, the CEREBEL trial failed to show any difference between the combination of capecitabine plus lapatinib and capecitabine plus trastuzumab in preventing metastatic spread to the brain [61].

The irreversible inhibitor neratinib was tested in the phase III NALA trial. The combination of neratinib plus capecitabine showed superior PFS, fewer interventions for CNS disease (surgery or radiotherapy) and an objective response rate (ORR) of 26.3% when compared with the standard of care (lapatinib plus capecitabine) in patients with metastatic disease previously treated with two or more regimens of anti-HER2 therapy. Importantly, patients could have symptomatic or stable BMs at study entry [17].

Tucatinib, a reversible HER2 inhibitor, was tested in the phase III HER2CLIMB study. The trial explored the combination of tucatinib, capecitabine and trastuzumab in mBC patients with BMs previously treated with pertuzumab, trastuzumab and trastuzumab emtansine (T-DM1). Patients with both stable and unstable BMs were allowed unless a local intervention was urgently needed.

Considering the population with BMs, the use of tucatinib, capecitabine and trastuzumab demonstrated better median PFS than placebo plus capecitabine and trastuzumab (7.6 and 5.4 months, respectively). Importantly, intracranial PFS and OS reached 9.9 and 18.1 months, respectively, whereas the ORR was 47.3% among patients with measurable BMs [18]. Based on these results, tucatinib is now approved in combination with capecitabine and trastuzumab in patients with HER2-positive metastatic breast cancer who have received one or more prior anti-HER2-based regimens in the metastatic setting, even in cases of BMs.

Recently, in the randomized phase III HER2CLIMB-02 study, the combination of tucatinib plus T-DM1 was compared to T-DM1 plus placebo in patients pretreated with trastuzumab and taxane. Patients with stable, progressing and untreated BMs were allowed. The combination significantly prolonged the median PFS both in the overall population and in patients with BMs (7.8 vs. 5.7 months) [62].

Turning to antibody–drug conjugates (ADCs), thus far, two drugs have been approved for the treatment of HER2-positive mBC.

The previously cited T-DM1 was the first approved ADC, which consists of the humanized monoclonal antibody trastuzumab with the payload emtansine, a microtubule inhibitory agent. However, no specific trial has explored T-DM1 in the treatment of BMs. Data were derived from post hoc and retrospective analyses from randomized trials.

In the retrospective exploratory analysis of the phase III EMILIA trial, T-DM1 significantly increased OS in patients with asymptomatic CNS metastases at baseline compared with the combination of capecitabine and lapatinib [63].

In addition, a real-world study conducted in a similar population [64] demonstrated a median PFS and OS of 7 and 14 months, respectively.

Finally, the exploratory analysis of the phase IIIb Kamilla trial conducted in pretreated mBC patients confirmed the activity of T-DM1 in 126 patients with BMs at baseline. T-DM1 led to an intracranial response rate of 42.9% and mPFS and OS rates of 6 and 19 months, respectively [14].

Trastuzumab deruxtecan (T-DXd) is the second innovative ADC consisting of a an anti-HER2 antibody, a cleavable tetrapeptide base as a reversible linker and a cytotoxic topoisomerase I inhibitor as the payload. The drug has a higher drug-to-antibody ratio than T-DM1 (8:1 vs. 4:1, respectively), and the released payload can cross the cell membrane, allowing the killing of neighboring cancer cells regardless of HER2 expression (by-stander effect). T-DXd has been recently approved as a second-line treatment of HER2+ metastatic breast cancer based on the results of DESTINY-Breast 03 trial. T-DXd demonstrated superiority over T-DM1 in terms both of PFS (29 vs. 7.2 months) and OS (52.6 vs. 42.7 months) in patients previously treated with a taxane and pertuzumab [65,66] across all subgroups, including patients with BMs.

Very recently, the exploratory DESTINY-Breast 01/-02/-03 analysis [67] highlighted the activity and efficacy of T-DXd in patients with both stable and active BMs.

The phase II DEBBRAH trial enrolled patients with a history of BMs, including leptomeningeal carcinomatosis, showing an ORR of 44.4% in patients with progression after local treatments (cohort 3) [68]. In addition, the small phase II TUXEDO-1 study evaluated patients with active BMs. Eleven out of fifteen enrolled patients exhibited intracranial ORRs as high as 73.3%. [69].

In the retrospective ROSET-BM trial, patients with HER2-positive mBC and symptomatic and asymptomatic BMs and those with stable and active BMs or leptomeningeal disease were treated with T-DXd [70]. The intracranial ORR in the 51 patients with brain lesion imaging data was 62.7%.

Finally, the results of the DESTINY-Breast 12 trial have been recently published [71]. Two-hundred and sixty-three patients with stable and active BMs were enrolled. Overall, in this cohort, the CNS PFS rates at 12 months were 58.9% and 60.1% in patients with stable and active BMs, respectively. In terms of activity, the overall CNS ORR was 71.7%, while values of 79.2% and 62.3% were reached in patients with stable and active BMs, respectively.

### 5.2. Triple-Negative Breast Cancer

Nearly 40% of patients with triple-negative breast cancer will develop BMs during their history [72]. Furthermore, it was described that germline mutations in BRCA1 or BRCA2 are associated with an increased risk of developing BMs, thus becoming a real issue considering that approximately 15 to 25% of all TNBC patients harbor germline BRCA1 or BRCA2 mutations [73,74]. Compared with those of other subtypes of BC, the survival time of TNBC patients after a diagnosis of BMs is shorter, with the majority of patients (60.5%) dying within 12 months after diagnosis [75].

The management of patients affected by BM, which includes local and systemic treatments, must be discussed within a multidisciplinary team. The classical approach includes a combination of systemic therapies after the administration of local therapy (surgery or radiotherapy) whenever progressive brain disease or symptomatic disease occurs. However, in cases in which local therapy is not indicated (due to the status of extracranial disease, patient conditions or comorbidities), systemic treatments alone should be proposed to control existing brain metastases together with extracranial disease to improve patients’ survival. Accordingly, a single-institution cohort study of 119 patients with BMs demonstrated that the use of systemic therapy after intracranial recurrence was a predictor of survival [75].

Historically, due to lack of specific therapeutic targets, chemotherapy has represented the major treatment for TNBC. However, the efficacy of both single-agent and combination chemotherapy is limited. Furthermore, solid data regarding the efficacy of chemotherapy in this setting are lacking. Chemotherapy drugs that can cross the BBB include capecitabine alone, the combination of temozolamide and capecitabine, platinum compounds and eribulin [76,77,78,79].

Capecitabine has shown activity in patients affected by BC with BMs with a tolerable safety profile. A retrospective single-center study reported a CNS ORR of 48% and a CNS disease control rate of 60% with capecitabine. A retrospective trial of 873 patients affected by BMs reported a median OS with capecitabine alone of 11.8 months [80].

Temozolamide, a drug used in clinical practice in the treatment of glioma, has the ability to cross the BBB. Data from the literature on its efficacy in BC are conflicting. A phase I study that evaluated temozolamide in combination with capecitabine reported an 18% ORR in women with BC and brain metastases. The treatment was well tolerated; the most common side effects were fatigue and nausea [77].

A systematic review of 14 clinical trials reported no advantages of concurrent whole-brain radiotherapy (WBRT) and temozolamide for BMs from BC [81]. Those results were confirmed by a phase II trial that enrolled 100 patients who received WBRT with or without temozolamide. WBRT combined with temozolamide did not significantly improve local disease control and survival. At 6 weeks, ORRs reached 36% in the WBRT arm and 30% in the WBRT + TMZ arm. The median PFS and OS were 7.4 and 11.1 months in the WBRT-alone arm and 6.9 and 9.4 months in the WBRT plus temozolomide arm, respectively [82].

Several trials evaluated combination treatments including bevacizumab. In a phase II trial, 38 patients received this anti-angiogenic agent in combination with carboplatin. The CNS ORR was 63%, suggesting the efficacy of this combination regimen [83].

A single-arm phase II study evaluated 35 BC patients with BMs refractory to WBRT who were treated with a 21-day cycle of bevacizumab followed by etoposide and cisplatin. The CNS ORR was 75% (95% CI, 42.8–94.5), and the median CNS progression-free survival (PFS) was 6.6 months (95% CI, 0.8–12.4), suggesting the efficacy of the combination in this setting of disease [84].

Considering platinum compounds, the combination of cisplatin and etoposide was evaluated in a large prospective trial including patients affected by lung cancer, melanoma and BC. The results showed a response rate of 37.5% in patients affected by BC [85].

Data regarding the efficacy of eribulin, a microtubule inhibitor that results in mitotic arrest and apoptosis, are poor and only retrospective. A retrospective observational study that evaluated the efficacy of eribulin in 20 patients with BMs reported a median CNS PFS of 3.39 months. Cox univariate analysis identified molecular subtypes as prognostic factors, with patients with TNBC being more likely to experience CNS progression than patients with luminal tumors (HR = 0.23 (95% CI, 0.07–0.80), *p* = 0.021) [86].

Furthermore, a novel treatment that yielded interesting results is liposomal irinotecan (Na-IRI). Na-IRI has the ability to cross the BBB and can cause higher tumor growth inhibition compared to irinotecan [87]. In a phase I trial, Na-IRI showed responses in patients with TNBC-induced BMs, with eight patients achieving stable disease in the CNS at 16 weeks. The most frequently encountered toxicities were diarrhea (grade 3 or higher diarrhea reported in 30% of patients), hypokalemia, hypophosphatemia and anemia [88].

Preclinical studies using a mouse model of breast cancer-induced brain metastases demonstrated improved survival with tirinotecan pegol (NKTR 102), a topoisomerase I inhibitor, compared to conventional irinotecan. [89]. The phase III BEACON trial showed a significant survival advantage for breast cancer patients with brain metastases receiving NKTR-102 treatment compared to treatments selected by physicians (median OS rates of 10.0 and 4.8 months, respectively) [90]. The ATTAIN phase III trial investigated the efficacy of etirinotecan pegol compared to treatments selected by physicians in patients with metastatic breast cancer, including those with stable brain metastases. The results were not consistent with the positive outcomes shown in the BEACON trial. In fact, no statistically significant differences in PFS and OS between treatment with etirinotecan pegol and chemotherapy were reported [91].

Also, in the TNBC setting ADCs have rapidly changed the therapeutic landscape of metastatic breast cancer. The DESTINY-Breast 04 trial enrolled patients with metastatic HER2-low BC to receive T-DXd versus standard chemotherapy. The results showed significantly longer PFS and OS for patients treated with T-DXd versus the standard of care, leading to FDA approval in 2022. Approximately 11% of the patients in this trial were hormone receptor-negative, and 6% of the patients treated with T-DxD had stable BMs. Adverse events of grade 3 or higher were reported in 52.6% of the patients who received trastuzumab deruxtecan; in particular, drug-related interstitial lung disease or pneumonitis occurred in 12.1% of the patients who received trastuzumab deruxtecan [92].

The DEBBRAH trial is a five-cohort, phase II study that enrolled patients with pretreated HER2-positive or HER2-low metastatic breast cancer with stable, untreated or progressing BMs and/or leptomeningeal carcinomatosis. Results regarding the two cohorts, which included patients with HER2-low BC, are not yet available. Early findings show a significant intracranial ORR of approximately 50%, suggesting the potential efficacy of this treatment [68].

Sacituzumab govitecan (SG) is a TROP2-targeting ADC drug approved for the treatment of metastatic TNBC according to the ASCENT trial’s results [20]. In this trial, PFS and OS were significantly longer with SG than with single-agent chemotherapy. In addition, 11% of the patients included in the study had stable BMs. In this subgroup of patients, SG showed a benefit compared to chemotherapy in terms of tumor response (with a clinical benefit rate of 9.4% versus 3.4%, respectively) and PFS (2.8 months for SG versus 1.6 months for chemotherapy, respectively). The most common adverse event related to sacituzumab govitecan was neutropenia (51% of patients), followed by leukopenia, diarrhea, anemia and febrile neutropenia [93].

PARP inhibitors are a targeted therapy that specifically targets the DNA damage response in BRCA1/2-mutated BC and has radically changed clinical practice in the treatment of this population.

In the phase III EMBRACA trial, talazoparib showed a benefit in terms of PFS and ORR versus chemotherapy. The benefit to PFS was even higher in the subgroup of BM patients (HR 0.32 vs. HR 0.58). Hematologic grade 3–4 adverse events (primarily anemia) occurred in 55% of the patients who received talazoparib, while non-hematologic grade 3 adverse events occurred in 32% of the patients [94]. In addition, olaparib also showed efficacy in TNBC based on the results of the OlympiAD trial. A post hoc analysis confirmed the benefit of olaparib in the small number of patients with BMs, with ORRs of 64% and 20% in the olaparib arm and the standard-of-care arm, respectively. The rates of grade 3 or higher adverse events (in particular, anemia, fatigue and gastrointestinal toxicities) were 36.6% in the olaparib group and 50.5% in the standard therapy group [95].

Safety and efficacy data are lacking for the use of immunotherapy in patients with BMs. The Keynote-355 trial, which showed the efficacy of pembrolizumab in metastatic TNBC, excluded patients with BMs [96]. Many clinical trials evaluating combination therapies with immunotherapy in patients with BMs are ongoing.

## 6. Future Perspectives: Past Trials and New Ongoing Trials

A total of nine trials were found to be eligible for the ongoing trial section, and they comprised both new experimental drugs and unexplored combinations of chemotherapies. The clinical trial characteristics are summarized in Table 1. For a comprehensive overview, we also collected data from completed and interrupted phase I and phase II trials, which are summarized in Table 2 and Table 3, respectively.

Among the drugs studied in the CNS, monoclonal antibodies targeting the vascular endothelial growth factor (VEGF) pathway showed good efficacy in enhancing PFS but not OS in glioblastoma [103], and it has often been proposed in BC as a first-line treatment, particularly in TNBC [104]. A small study was conducted in BC patients affected by leptomeningeal BC metastases from HER2+ and TNBC subtypes, showing a low rate of survival of 4.7 months [97] for both neurological PFS and OS. These results were disappointing due to survival outcomes largely comparable to those with other systemic therapies administered in this setting [102], although the administration of bevacizumab resulted in a reasonable reduction in peripheral vascularization of BMs [105]. However, a direct placebo-controlled comparison study should be conducted for definitive conclusions. Notably, the combination of bevacizumab with cisplatin and etoposide (BEEP scheme) significantly increased survival to 9.7 months compared to 1.7 months with non-BEEP regimens in a recent retrospective study [106]. However, the study had a small sample size and reported selection and performance biases (lacking randomization and masking procedures); for this reason, a phase III study is needed.

Among ongoing clinical trials, a promising treatment is based on chimeric antigen receptor T (CAR-T). CAR-T technology has revolutionized the treatment landscape for hematological malignancies, but recently, a strong interest in reshaping cytotoxic immunity responses in solid tumors has emerged in anergic tumoral contexts [107]. This technology consists of modulating T-cell lymphocyte activation through genetic engineering of T-cell receptors (TCRs) for recognition of tumoral cells [108]. A currently ongoing phase I trial is investigating TCR-engineered cells’ ability to find and kill HER2+ tumoral cells (HER2-CAR-T cells) and searching for the best doses for CAR-T administration to limit toxicities in patient affected by CNS and leptomeningeal metastases from HER2+ BC (NCT03696030). Preclinical studies confirmed that CAR-T cells were able to pass through the BBB, with interesting consequences for BC-induced BM treatments despite a significant increase in severe CNS side effects, such as the deadly “immune effector cell-associated neurotoxicity syndrome” (ICANS) [108]. Despite the possible severe autoimmunities described, CAR-T cells are very promising in this poor prognosis subgroup. However, only one study was retrieved from our query on clinicaltrials.gov, which focused on HER2+ disease, whereas studies on CAR-T cells used in TNBC with CNS metastases were not found.

A new investigational drug, the experimental drug ANG1005, was compared with the actual best standard-of-care therapies in patients affected by HER2- BC with stable CNS metastases previously treated with encephalic radiotherapy in the randomized phase III ANGLeD trial (NCT03613181). ANG1005 is a taxane-conjugated peptide (paclitaxel trevatide) in which three molecules of taxol are covalently linked to a proteic core, angiopep-2, greatly enhancing paclitaxel‘s capacity to pass through the BBB and blood–cerebrospinal barriers through LRP1 transport proteins [109,110]. Accordingly, preclinical studies showed that ANG1005 had 86-fold greater CNS diffusivity than regular paclitaxel [111], and clinical evidence of its efficacy has been previously shown in a phase II trial [109]. In this study, the subgroup with leptomeningeal metastases highlighted a 63% OS rate at 8 months with ANG1005 treatment, largely surpassing the median OS of 4 months described in literature for standard therapies [109]. A possible side effect that can be concerning from this greater accumulation of paclitaxel in the CNS and BMs is central neurotoxicity. Although this effect could be a putative toxicity, it was not described in the previous phase II study [107]. In addition, in a previous phase I of the same drug administered in the setting of glioblastoma multiforme, the authors performed neurocognitive tests, asserting the apparent absence of central neurotoxicity [109]. Nevertheless, the results from the ANGLeD trial will be pivotal for confirming this important survival gain.

In addition to the systemic route, another important experimental direction in the setting of CNS metastases is other administration routes. Intrathecal administration is an interesting route that has been investigated in several past clinical trials [98,99,100,101,102]. Currently ongoing clinical trials in the setting of HER2+ BC-induced BM are based on the intrathecal administration of drugs such as pertuzumab/trastuzumab after brain radiotherapy (NCT04588545), intrathecal thiotepa in combination with methotrexate (NCT06543992) and trastuzumab/capecitabine/tucatinib (NCT05800275). HER2 agents show good but heterogeneous and unpredictable penetration through the BBB [112]. In fact, despite these important pieces of evidence, a recent metanalysis revealed that intrathecal administration of trastuzumab alone yielded no benefits for OS and CNS-specific-PFS [113]. However, combinations with dual blockade have not yet been evaluated, although stably high drug concentrations can be reached through administration into the cerebrospinal fluid, maximizing the effect in patients with leptomeningeal localizations. However, NCT04588545 trial results are expected in the early 2025. Concerning thiotepa, it is a trifunctional alkylating drug used in hematological malignancies and bladder, breast and ovarian cancers [114]. Its intrathecal use in high doses has been evaluated in leptomeningeal localizations from breast cancer, revealing a median survival of 4.5 months when used in first or second lines of therapy [98], which is largely comparable to rates with other previously cited treatments in the literature [106]. This result was even less pronounced when intrathecal thiotepa was used in patients from the third line onwards, with a significant negative prognostic factor described based on the multivariate analysis, suggesting that poly-treated patients develop primary resistance to thiotepa [98]. Considering that the spread of breast cancer cells to leptomeningeal localizations is a late outcome and patients are usually already heavily treated at this point, this study describes another disappointing result. On the other hand, intrathecal methotrexate is a drug largely used in this scenario but with poor results [99,100,101,115], particularly in pretreatment settings [106]. However, the combination of thiotepa and methotrexate has not yet been explored, although the combination of an alkylating factor (thiotepa) with an anti-metabolic cytostatic agent (methotrexate, a metyl-tetra-folate reductase inhibitor) provided a scientifically sound rationale for the NCT06543992 trial. Despite some skepticism, we expect the results of this trial in late 2026. Trastuzumab, capecitabine and tucatinib in combination, as described previously, were tested as a systemic therapy for a second or third line of treatment in the HER2CLIMB trial, proving its efficacy in previously untreated symptomatic BC-induced BMs [18]. Although good penetration of the BBB was seen for tucatinib and capecitabine, intrathecal administration could improve the treatment of patients with brain-prevalent and leptomeningeal disease.

A new experimental drug, the compound QBS72S, will be investigated in a single-arm phase II trial evaluating efficacy and safety (NCT05305365). This new compound combines a cytotoxic effect conferred by the tertiary N-bis(2-chloroethyl)amine with good brain diffusion through the BBB due to the presence of the large-amino acid-transporter 1 (LAT1) substrate. After administration, the binding between QBS10072S and the transporter LAT1 allows passage through the BBB and entry inside LAT1-expressing cells. After entry into tumoral cells, the cytotoxic domain binds to DNA, causing cell cycle arrest and inducing apoptosis [116]. The LAT1 protein is expressed largely in brain metastases, particularly in those from TNBC, whereas lower levels are expressed in normal brain tissue, allowing QBS10072S to preferentially target brain tumors [117]. The results of this innovative phase IIa proof-of-concept trial are expected in early 2025.

Concerning experimental drugs, the TUXEDO-3 phase II trial (NCT05865990) is a very promising investigation in the setting of BC-induced BMs involving patritumab–deruxtecan (HER3-DXd), an anti-HER3 ADC. Similar to TUXEDO-1, which focused on T-DXd efficacy in the setting of BC-induced BMs [69], TUXEDO-3 will investigate the use of HER3-DXd in the setting of active brain and leptomeningeal metastases not only from BC but also from lung cancer; indeed, HER3-DXd has been previously used in several settings like metastatic lung cancer [116,118] and metastatic TNBC [119]. Interestingly, positive results came from a phase I/II trial where HER3-DXd was used in heavily pretreated mBC patients, showing better outcomes in HER2+ patients than in TNBC and HR+/HER2- patients. In particular, in extracranially prevalent disease, patients with HER2+ breast cancer (n = 14) showed the best outcomes, with an ORR of 42.9% and mPFS of 11.0 months, whereas HR+/HER2- BC (n = 113) and TNBC (n = 53) patients showed rates of 30.1% and 7.4 months and 22.6% and 5.5 months, respectively. Notably, the efficacy of the drug was independent of HER3 levels, with comparable outcomes with both high and low expression levels [120]. Although the sample size was very limited, particularly in the HER2+ setting, the results were very promising when compared with the survival benefit of 4 months for standard therapies. However, this should not be surprising considering the extensive evidence demonstrating HER3 as a resistance factor for HER2 pathways, increasing in HER2 heavily pretreated patients and enhancing tumoral survival [121,122,123]. However, for definitive results from TUXEDO-3, we have to wait until early 2026.

Another interesting experimental ADC, datopotomab–deruxtecan (DATO-DXd), will be investigated in the DATO-BASE phase II trial (NCT06176261), an open-label, single-arm study that is enrolling HER2- BC patients and distributing them into three different cohorts for different settings (HR+/HER2-, TNBC and leptomeningeal metastases). DATO-DXd has been recently studied in a phase I trial in metastatic HR+/HER2- BC and TNBC, showing a good safety profile and admissibility for further studies. However, for data on BC-induced BM responses and survival, we will have to wait until mid-2028. A general overview of these old and new strategies is graphically summarized in Figure 2.

## 7. Conclusions

Breast cancer can eventually develop brain metastases in 40–50% of cases in HER2-positive BC and TNBC, and this risk can further increase in cases of BRCA1/2-mutated patients affected by TNBC. Importantly, this setting is still burdened by a very poor prognosis with a median overall survival of 4–5 months as described in literature. Basic and translational studies, which shed light on the mechanisms underlying this complex process of metastatization, are essential, and further studies will be needed to delineate the complex interplay between brain metastatic cells and the tumoral niche, particularly microglia, astrocytes and pericytes, which can not only recognize and destroy breast cancer cells but are also responsible for cell growth and cell survival if recruited and re-educated in the tumoral niche. In the HER2 setting, antibody–drug-conjugates such as T-DXd are innovative and promising tools with increasing evidence, particularly in unstable and progressing brain metastases, whereas their use in HER2-low TNBC has limited data, and mature results from the DEBBRAH trial are anticipated for this purpose. Similarly, TROP2-targeting antibody–drug conjugates need further studies to prove a real benefit in this subgroup, although an increased ORR was observed. Historically, studies have systematically excluded leptomeningeal metastases and unstable localizations. Remarkably, similarly to T-DXd and tucatinib, the ongoing trials that we presented show an increased interest in these important subgroups, with the aim to answer to an actual issue. However, although this review focused on systemic therapies, we should not forget that local therapies are available and commonly prescribed in this setting, such as stereotactic radiotherapy or radiosurgery (gamma-knife radiotherapy), which could positively impact patient survival. Considering that stabilization of unstable intracranial disease is based on these local therapies, it will be pivotal in the future to ascertain if the best strategy could be the old trend (administering brain radiotherapy and then systemic treatment) or if new therapies can delay radiotherapy in further lines of treatment showing possible increases in survival or delay radiotherapy side effects.

## Figures and Tables

**Figure 1 cancers-16-04164-f001:**
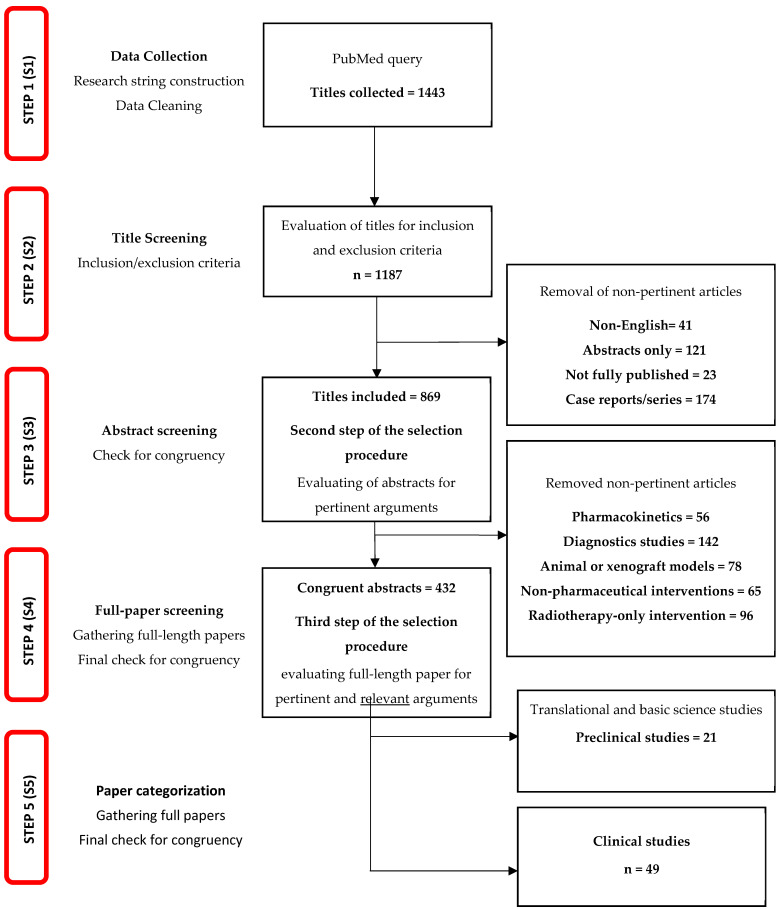
Flowchart of the selected studies (for details see methods).

**Figure 2 cancers-16-04164-f002:**
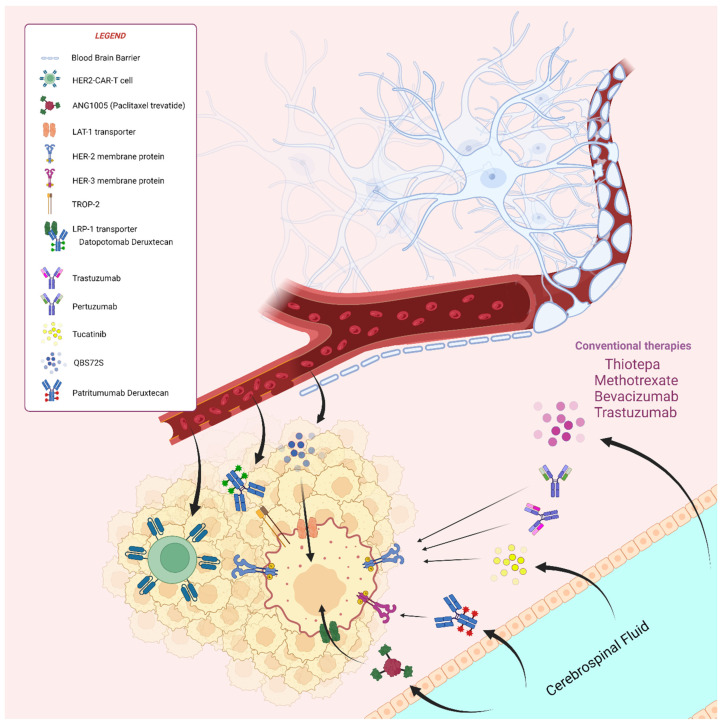
Overview of old and new treatment strategies in the management of brain metastases from breast cancer. Image created with BioRender.com by Roberto Rosenfeld. Copyright must be credited to the authors of this review.

**Table 1 cancers-16-04164-t001:** Ongoing trials for leptomeningeal and brain metastases.

NCT ID	Investigational Drug	Setting	Clinical Phase	Expected Enrolment	Randomization	Masking	Study Start Year	Primary Completion
NCT03696030	HER2 CAR-T cells	HER2+ mBC	Phase I	39	No	No	2018	Early 2025
NCT03613181	ANG1005	HER2- mBC	Phase III	150	Yes	No (open label)	2023	Late 2024
NCT05305365	QBS72S	Undefined mBC	Phase IIa	40	No	No (open label)	2022	Early2026
NCT05865990	HER3-DXd	HER2- mBC	Phase II	60	Not applicable (single arm)	No	2023	Late 2025
NCT06176261	Datopotomab≥ deruxtecan	HER2- mBC	Phase II	58	No	No (open label)	2023	Early 2028
NCT04588545	Intrathecal TTZ/PTZ in post-RT treatment	HER2+ mBC	Phase I/II	39	No	No	2020	Late 2024
NCT06543992	Intrathecal thiotepa	Not defined mBC	Phase II	22	No (single-arm study)	No (open label)	2024	Late 2026
NCT05800275	Capecitabine, tucatinib and intrathecal trastuzumab	HER2+ mBC	Phase II	30	No (single-arm study)	No (open label)	2023	Late 2026
NCT02422641	High-dose MTX	Undefined mBC	Phase II	16	No (single-arm study)	No (open label)	2015	Late October 2025

HER3-DXd: Patritumab–deruxtecan; mBC: metastatic breast cancer.

**Table 2 cancers-16-04164-t002:** Completed trials involving investigational drugs or schemes for CNS metastases from breast cancer.

Author (Year)	Investigational Drug	Study Design	OS	Neurologic PFS	Toxicities, G3/G4
Wu PF et al. (2015) [97]	CDDP+ etoposide + bevacizumab	Phase II Single arm study	4.7 mos	4.7 mos	Neutropenia (23.1%), leukopenia (23.1%) and hyponatremia (23.1%)
Comte A et al. (2013) [98]	Intrahecal thiotepa	Phase II	4.5 mos	-	Not assessed; no patient discontinued the treatment for toxicity
Pan Z et al. (2016) [99]	Intrathecal methotrexate	Phase II	6.5 mos	4.0 mos	Encephalitis 2%, radiculitis 13%
Niwińska A et al. (2014) [100]	Intrathecal methotrexate vs. ctytarabine	Phase II	4.2 mos	-	Neurotoxicity
Glantz MJ et al. (1999) [101]	DepoCyt (cytarabine)	Phase II	3.3 mos	1.9 mos	Comparable toxicities with methotrexate
Le Rhun et al. (2020) [102]	Lyposomal cytarabine	Phase II	7.0 mos	3.8 mos	Psychiatric, myelotoxicity, neurotoxicity and gastrointestinal

**Table 3 cancers-16-04164-t003:** Interrupted trials.

NCT ID	Investigational Drug	Setting	Clinical Phase	Study Design	Status	Study Start Year	Year of Interruption/Withdrawn	Reason
NCT04856475	Neratinib	BMs in advanced HER2+ BC	Phase II	Open-label, non-randomized	Withdrawn	Never started	24 November 2021	Termination of collaboration with PUMA
NCT03661424	Bispecific antibody-armed T cells (BATs)	LMs in mBC	Phase I	Dose-finding, open-label, non-randomized	Terminated	2019-02-26	14 December 2021	Slow study accrual, partially due to the pandemic
NCT06137651	Trotabresib (combined with VNR and RT)	BMs in advanced HER2+ BC	Phase I	Dose-finding, open-label, non-randomized	Withdrawn	Never started	28 March 2024	Study terminated by the sponsor

BC: breast cancer; BMs: brain metastases; LM: leptomeningeal metastases; RT: radiotherapy; VNR: vinorelbine.

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
