# Peer review of "Current Evidence in the Systemic Treatment of Brain Metastases from Breast Cancer and Future Perspectives on New Drugs, Combinations and Administration Routes: A Narrative Review"

_cancers, 2024, doi:10.3390/cancers16244164_

Round 1

Reviewer 1 Report

Comments and Suggestions for Authors

This review entitled “Current evidence in the systemic treatment of brain metastases from breast cancer. A narrative review.” covers various therapeutic options for breast cancer with brain metastasis. The authors discuss multiple important aspects of the challenges of the treatment including current ongoing clinical trials. The presentation, however, is fragmented and not well laid out in order. Reorganizing the flow of narrative and language editing would enhance the quality of the manuscript.

One of the problems that contribute to the fragmented narrative flow is there are too many very short paragraphs consisting of just one or two sentences.

Figure 2. Recently it is reported that neurons can make synapse with metastatic tumor cells including breast cancer cells and change the tumor biology. Microglia also interacts with metastatic tumor as the authors Including these missing cells in the Figure 2 may better reflects microenvironment of the brain metastatic tumor. Authors can discuss potential therapeutic approaches targeting such neuron-tumor synapses

The authors discussed not only systemic treatment approach but also localized approaches such as intrathecal administration. So, the wording of the title, "the systemic treatment of brain..." may not reflecting what is discussed in this review manuscript.

Numbering of the section requires correction. Both "Biomolecular features"and "Systemic treatment" sections have the same number 4.

Line 60, "On the other hand" This phrase is usually used with the phrase, 'on one hand' to indicate two things are contrasting each other.

On lines 102 to 105, the authors listed examples of non-pertinet publication eliminated from the study. Why did the authors judge blood brain barrier permeability is not pertinent? This barrier is a critical factor to treat brain metastasis by systemic drug administration. The authors also point out the route of administration is important (lines 464 and thereafter,) discussing about intrathecal administration.

Lines 139-140; The authors wrote, "for this reason highly effective drugs were recently developed, improving the outcomes in these settings." The authors need to provide more information on the "effective drugs", and how they improved the outcomes.

Lines 204-208; The authors discuss on cancer stem cells. Throughout the manuscript, no definition nor biological, clinical meaning of cancer stem cells, which makes it difficult to understand the discussion points presented here.

On line 255, the abbreviation, "T-DM1" appears without indication of full terminology, which appears later in line 267. Organizing the order of presentation would help readers' comprehension. Also, making an abbreviation list would be helful.

Lines 452-463; Taxane-conjugated peptide penetrates blood brain barrier better than paclitaxel. This might be beneficial for targeting brain metastasized breast cancer cells, but higher concentration of taxane may cause central neurotoxicity. Do the authors have any thoughts on this?

Comments on the Quality of English Language

There are too many too-short paragraphs, which block the flow of narrative.

Reviewer 2 Report

Comments and Suggestions for Authors

This paper provides a detailed review of the current status of breast cancer brain metastasis.

Appropriate procedures were followed to determine the review subjects, and a vast number of papers were reviewed to provide an overview of treatment.

Points to note

This paper is about systemic treatment, and it is certainly a review of systemic treatment for breast cancer brain metastasis. However, there are also reports that treatments other than systemic therapy, such as stereotactic radiosurgery, can extend survival time. To prevent readers from misreading the conclusion, I think the text should also briefly mention treatments other than systemic therapy.

Author Response

Reviewer #2

This paper provides a detailed review of the current status of breast cancer brain metastasis.

Appropriate procedures were followed to determine the review subjects, and a vast number of papers were reviewed to provide an overview of treatment.

Points to note

This paper is about systemic treatment, and it is certainly a review of systemic treatment for breast cancer brain metastasis. However, there are also reports that treatments other than systemic therapy, such as stereotactic radiosurgery, can extend survival time. To prevent readers from misreading the conclusion, I think the text should also briefly mention treatments other than systemic therapy.

We thank warmly the reviewer for the suggestion. We implemented the conclusion in order to be more comprehensive and adding some consideration for future perspectives.

Reviewer 3 Report

Comments and Suggestions for Authors

This is an interesting review of the literature and up to date about the spectrum of treatment of the breast cancer with a particular attention to the HER2 positive and TNBC subgroups which demonstrated high percentages of brain metastasis among all others. The topic is discussed in detail, the review structure is good same as the selection of papers. I suggest the article for publication.

Author Response

We thank the reviewers for their comments on our work.

Reviewer 4 Report

Comments and Suggestions for Authors

The review paper, "Current evidence in the systemic treatment of brain metastases from breast cancer," by Garonne et al, discusses he newest therapeutic modalities and challenges in managing brain metastases from breast cancer.

It is comprehensive, reviews systematically existing and experimental therapies and gives emphasis on molecular characteristics and the microenvironment in treatment response. However, some sections lack granular discussion on drug safety profiles and treatment options for TNBC are limited and could be explored further.

Author Response

The review paper, "Current evidence in the systemic treatment of brain metastases from breast cancer," by Garonne et al, discusses he newest therapeutic modalities and challenges in managing brain metastases from breast cancer.

It is comprehensive, reviews systematically existing and experimental therapies and gives emphasis on molecular characteristics and the microenvironment in treatment response. However, some sections lack granular discussion on drug safety profiles and treatment options for TNBC are limited and could be explored further.

We thank the reviewer for this suggestion.

Accordingly, we added drugs safety profiles and further treatment options for TNBC patients.

Round 2

Reviewer 1 Report

Comments and Suggestions for Authors

Interaction between breast cancer cells and neurons may play critical roles in pathophysiology of the breast cancer brain metastasis, as exemplified by the recent reports [10.1101/2024.01.08.574608, 10.1038/s41586-019-1576-6], although this reviewer is not aware of any studies in guinea pigs mentioned by the authors. The authors declined to discuss this emerging topic of importance by saying " all the authors decided collegially to include only clinical trials on humans and therefore implementing only phase I-III trials," however the biology of neuron-breast cancer cell interaction would be worth being discussed in this review as the authors wisely expressed the importance of the knowledge on mechanism saying "the knowledge of the molecular mechanisms leading to the survival and growth of cancer cells within the brain may identify actionable target for new therapy" (lines 152-154). In addition, this neuron-breast cancer cell interaction is one of the components of microenvironment of brain metastasis. Discussing this topic in the section 4 would strengthen the comprehensive nature of this review. From this perspective, it is a thought to add neurons, and other types of cells discussed in this review (microglia, pericytes) to the Figure 2 to make it better depicting the microenvironment of the brain metastatic breast cancer.

Figure 2 is not mentioned in the main text. Narrative explanation in the main text or adding figure legend would help understanding of the readers. The caption for the Figure 2 describes "old and new systemic treatment". Aren't some of the reagents administered not systemically? Do authors intend to show that the vessel close to the tumor has broken blood brain varier by showing the blood vessel part unattached by astrocytes?

The phrase, "On the other hand", was replaced with "In addition,". This change made the phrase, "in addition", appears twice in a single sentence. More thorough English editing would enhance the quality of the manuscript.

Regarding exclusion of studies on BBB, the authors wrote in their response letter, "These studies on BBB permeability are about Physiology and Pathophysiology of brain metastases, including analyses on oncotic pressure and diffusion of new or old compounds/drugs, which are beyond the focus of this review." Including this reason in to the main text would minimize the readers' confusion.

In response to this reviewers's request to elaborate the biological or clinical significance of "cancer stem cell", the authors wrote as follows.

"The original lines 204-208 are now 239-243

We do not discuss about cancer stem cells. We simply point out that:

1) TNBC has the highest rate of stem cells

2) The connection between the protumor astrocytes activity and stem cells (see ref. 48)"

However, the lines 239-243 of the revised manuscript has nothing to do with cancer stem cell. The readers will benefit if the author provide a brief description on the biological significance of cancer stem cells in the context of brain metastasized breast cancer.

Comments on the Quality of English Language

Thorough language editing will improve the quality of the manuscript.

Round 3

Reviewer 1 Report

Comments and Suggestions for Authors

The authors nicely addressed the issues pointed out by this reviewer. Inclusion of neuro-tumor interraction and a brief description of cancer stem cell with well selected references will help readers who are new to this field to educate themselves further. Not citing the pre-print, although it is an intriguing report, is a respectable decision.

There is one minor change that the authors can make to further enhance the value of this review, which is to add a figure legend to Figure 2 explaining the leaky tumor blood vessel depicted by the lack of astrocytes coverage.

Author Response

Reviewer: There is one minor change that the authors can make to further enhance the value of this review, which is to add a figure legend to Figure 2 explaining the leaky tumor blood vessel depicted by the lack of astrocytes coverage.

We thank the reviewer for her/his suggestion. However such a legend is out of the scope of the figure 2 which is intended to give an overview of old and new treatment strategies in the management of brain metastases from breast cancer. In our opinion could be misleading for the readers